# Voxel based morphometry-detected white matter volume loss after multi-modality treatment in high grade glioma patients

Jesse D. de Groot[1], Bart R. J. van Dijken[1]*, Hiska L. van der Weide[2], Roelien H. Enting[3], Anouk van der Hoorn[1]

1 Department of Radiology, Medical Imaging Center (MIC), University Medical Center Groningen, University of Groningen, Groningen, The Netherlands, 2 Department of Radiation Oncology, University Medical Center Groningen, University of Groningen, Groningen, The Netherlands, 3 Department of Neurology, University Medical Center Groningen, University of Groningen, Groningen, The Netherlands

* b.r.j.van.dijken@umcg.nl

## Abstract

**Data Availability Statement:** Data will not be made available to other researchers as no patient approval has been obtained for sharing coded data. Syntax files will be made available from the Loket

### Background

Radiotherapy (RT) and chemotherapy are components of standard multi-modality treatment of high grade gliomas (HGG) aimed at achieving local tumor control. Treatment is neurotoxic and RT plays an important role in this, inducing damage even distant to the RT target volume.

### Purpose

This retrospective longitudinal study evaluated the effect of treatment on white matter and gray matter volume in the tumor-free hemisphere of HGG patients using voxel based morphometry (VBM).

### Method

3D T1-weighted MR images of 12 HGG patients at multiple timepoints during standard treatment were analyzed using VBM. Segmentation of white matter and gray matter of the tumor-free hemisphere was performed. Multiple general linear models were used to asses white matter and gray matter volumetric differences between time points. A mean RT dose map was created and compared to the VBM results.

### Results

Diffuse loss of white matter volume, mainly throughout the frontal and parietal lobe, was found, grossly overlapping regions that received the highest RT dose. Significant loss of white matter was first noticed after three cycles of chemotherapy and persisted after the completion of standard treatment. No significant loss of white matter volume was observed between pre-RT and the first post-RT follow-up timepoint, indicating a delayed effect.

Contract Research, section of the UMCG's Department of Legal Affairs, on reasonable request (email address: loket_contract_research@umcg.nl).

**Funding:** The author(s) received no specific funding for this work.

**Competing interests:** The authors have declared that no competing interests exist.

**Abbreviations:** CCRT, Concomitant chemoradiotherapy; DARTEL, Diffeomorphic anatomical registration through exponentiated lie algebra; DTI, Diffusion tensor imaging; HGG, High grade glioma; RT, Radiotherapy; SPM, Statistical parametric mapping; TMZ, Temozolomide; VBM, Voxel based morphometry.

## Conclusion

This study demonstrated diffuse and early-delayed decreases in white matter volume of the tumor-free hemisphere in HGG patients after standard treatment. White matter volume changes occurred mainly throughout the frontal and parietal lobe and grossly overlapped with areas that received the highest RT dose.

## Introduction

High-grade gliomas (HGG) are the most common form of malignant primary brain tumors, associated with a poor prognosis despite treatment [1, 2]. Radiotherapy (RT) and adjuvant chemotherapy are main elements in the treatment of HGG [3, 4]. After surgical resection of tumor, RT and chemotherapy aim at residual microscopic infiltrating tumor cells and are therefore both essential in achieving local control. A total radiation dose of 60 Gy, administered as 30 fractions of 2 Gy, is delivered at a target volume, consisting of the resection cavity or residual tumor plus a 1–2 cm margin. However, despite optimal planning, irradiation of macroscopic healthy brain tissue is inevitable, inducing RT-induced brain damage.

Previous MRI studies have shown that fractionated RT is able to cause structural changes of the white matter and gray matter of the brain, even outside and distant to the RT target volume [5–8]. These changes have been associated with neurocognitive decline which develops in nearly all HGG patients treated with RT [9, 10]. It is known that RT-induced damage to the hippocampus is associated with cognitive decline and is therefore generally spared [11]. Morphologic changes to other specific brain areas due to RT exposure are yet to be better understood.

Voxel Based Morphometry (VBM) analysis allows for voxel-wise comparison of tissue volume changes between different MRI timepoints for a population [12, 13]. VBM analysis has previously shown to be very useful in detecting longitudinal volume changes in glioma patients [14–16]. However if, and to what extent treatment affects healthy contralateral brain tissue during treatment, remains largely unknown. In this study, we evaluated the effect of standard treatment with RT and chemotherapy on gray and white matter volume of the tumor-free hemisphere of patients with unilateral HGG using VBM, aiming at the identification of the most vulnerable neuroanatomical brain areas.

## Methods

### Study population

We retrospectively included patients 18 to 80 years of age with histologically confirmed HGG, who underwent concomitant chemoradiotherapy (CCRT) with temozolomide (TMZ) followed by adjuvant chemotherapy according to the Stupp protocol [3] at the University Medical Center of Groningen (UMCG) between 2017 and 2020. Other inclusion criteria were acquisition of imaging follow-up after RT with at least a 3D T1-weighted MRI. Dexamethasone was allowed to be given as needed to control symptoms caused by cerebral edema. This resulted in the initial selection of 23 patients. The exclusion criteria were the following: the presence of HGG lesions in both hemispheres (N = 4), the presence of preexisting brain lesions or other brain abnormalities (N = 3), a history of prior brain surgery or irradiation (N = 0), different RT dosage (N = 1), missing MRI data (N = 0) or MRI data not conform with the study specific sequence or protocol (N = 3). The study was approved by the institutional review board and the need for written informed consent was waived.

## MRI data acquisition

All patients were scanned on a 1.5T Siemens Magnetom Aera scanner (Siemens Healthcare, Erlangen, Germany) with a 20-channel head coil between 2017 and 2020 in the UMCG. Multiple 3D T1-weighted sagittal scans at different timepoints were acquired of all patients: i) pre-RT, ii) post-CCRT, iii) after three cycles of TMZ and iv) after six cycles of TMZ. 3D T1 MP-RAGE images were acquired (repetition time [TR] 2200 ms, echo time [TE] 2.67 ms, inversion time [TI] 900 ms, field of view [FOV] 230 x 230 mm$^2$, matrix 256 x 256, slice thickness 1 mm, no spacing, resolution 0.96 mm$^3$, voxel size 1 x 0.977 x 0.977 mm$^3$, flip angle 8˚).

## RT planning

RT planning was performed on the reference CT image of each patient using Mirada RTx (Mirada medical, Oxford, UK). The RT planning technique used in all plans was volumetric modulated arc therapy (VMAT) with or without a static conformal non-coplanar beam. All patients received a total radiation dose of 60 Gy, administered as 30 fractions of 2 Gy daily during 6 weeks. The clinical target volume (CTV) consisted of the resection cavity and/or residual tumor plus a 1.5 cm margin without dose spillage to the contralateral hemisphere. Using the AI autocontouring function of Mirada RTx, the cerebrum was contoured on all CT images and then manually divided to acquire RT dose distribution data of the total cerebrum and of each hemisphere.

Furthermore, a mean RT dose map was created using 3D Slicer version 4.8.1 (http://www.slicer.org). Firstly, the maps of the RT dose distribution and corresponding reference CT images of each patient were acquired and selectively flipped to ensure the tumor-containing hemisphere was identical for all data. Secondly, all RT dose maps and CT images were coregistered using the general registration (Elastix) function in 3D Slicer after which a mean RT dose map for the cohort was created with the Radiotherapy Dose Accumulation function. Finally, the mean RT dose map was coregistered to the T1 weighted Montreal Neurological Institute (MNI) brain. In doing so, coordinates were transformed into a common coordinate space which allowed for identification of brain regions that received the highest RT dose.

## Image analysis

Image processing and voxel-based statistical analysis were conducted using Statistical Parametric Mapping (SPM), version 12 (Wellcome Department of Imaging Neuroscience Group, London, UK). Firstly, the T1-weighted images were reorientated with affine transformations to align these images with the anterior commissure–posterior commissure (AC-PC) line. Secondly, the images were selectively flipped to ensure the "tumor-containing" hemisphere was identical for all images. This was followed by applying the Diffeomorphic Anatomical Registration Through Exponentiated Lie algebra (DARTEL) tool to create a study specific template [17]. Segmentation into gray matter, white matter and cerebrospinal fluid was performed following standard procedures [18, 19]. Gray and white matter templates were used to normalize the gray and white matter images of each subject to the respective template; the resulting normalized images were modulated with a Jacobian correction; and an 8mm full width at half maximum Gaussian kernel was used to smooth all normalized and modulated images. The tumor-containing hemisphere was masked during all analyses as SPM is unable to correctly identify and segment gray matter and white matter in the proximity of tumors. Together with the selective flipping, this process ensures that the "tumor-free" hemisphere can be reliably analyzed [20].

## Statistical analysis of MRI data

Multiple general linear models were setup in SPM to asses gray and white matter volumetric differences between different time points. Post hoc, paired *t* tests were applied to compare gray and white volume between two distinct time points. To account for differences in brain size, the intracranial volume of each scan was calculated and used as a proportional correction factor (i.e. global normalization). VBM analysis was carried out with the voxel threshold set at uncorrected $p < 0.001$ ($p_{unc}$) and cluster threshold set according to $p < 0.05$ after family-wise error correction ($p_{FWEc}$).

## Results

### Patient characteristics

A total of 12 patients were included in this study with a median age at RT of 54.4 years (16.4 interquartile range [IQR], 45.6–62.0). See Table 1 for patient characteristics. Based on the diagnostic algorithm for the classification of gliomas in adults by the European council of neuro-oncology, a single grade II glioma patient was considered as having an HGG due to molecular markers [21]. The O(6)-methylguanine-DNA methyltransferase (MGMT) promotor status was known in 9/12 (75%) patients. MGMT was methylated in 6 patients (50%) and unmethylated in 3 patients (25%). The mean RT dose of the total cerebrum, tumor-containing hemisphere and tumor-free hemisphere were 25.2 (4.15 IQR, 24.4–28.6) Gy, 33.5 (6.38 IQR, 31.6–38.0) Gy and 17.1 (6.93 IQR, 13.5–20.4) Gy, respectively. The highest measured mean RT dose was 35.2 Gy. An average dose distribution map is shown in Fig 1 and demonstrates the highest mean RT dose (20–30 Gy) was delivered at the parietal and occipital lobes and thalamus, and a lower dose (10–20 Gy) was received in the frontal and temporal lobes and basal ganglia.

The preRT scan was on average 67.4 days (7.25 IQR, 65.3–72.5) before RT. The first post-RT scan was on average 21.8 days (4.75 IQR, 20.8–25.5) after RT. The second post-RT scan was on average 113 days (6.00 IQR, 105–111) after RT. The third post-RT scan was on average 201 days (11.5 IQR, 189–201) after RT. Two patients had no eligible MRI after six cycles of TMZ due to tumor progression causing the tumor to infiltrate over the midline. A total of 46 MR images were used in the analysis.

### White matter analysis

Voxel-wise comparison of white matter volume revealed significant clusters indicating white matter volume loss after radiotherapy (Table 2). Comparison of pre-RT images with post-RT images revealed no significant clusters for volumetric difference in white matter (Table 2). Two significant clusters were found for white matter volume loss, extending throughout the corpus callosum, cingulate gyri (anterior, middle and posterior gyrus) and the frontal (precentral and superior, middle and inferior gyrus), parietal (postcentral, supramarginal, angular and inferior parietal gyri), temporal (superior temporal gyrus) and occipital (lingual, calcarine, precuneus, cuneus, fusiform, superior and middle occipital gyrus) lobe, when comparing pre-RT images with the images after three cycles of TMZ (Fig 2A). Three significant clusters for white matter volume loss were found in similar areas with the addition of the insula, putamen, pallidum, superior parietal and inferior occipital gyrus when comparing pre-RT images and the images after 6 cycles of TMZ (Fig 2B).

Additionally, when comparing post-RT images with images after three cycles of TMZ, five significant clusters for white matter volume loss were found throughout the corpus callosum, putamen, pallidum, hippocampus, parahippocampal zone, insula, cingulate gyri (anterior, middle and posterior gyrus) and frontal (inferior gyrus), parietal (postcentral, supramarginal and inferior

**Table 1. Demographic and clinical parameters.**

| Parameter | All patients $n$ = 12 (%) |
|---|---:|
| *Gender* | |
| Male | 6 (50.0) |
| Female | 6 (50.0) |
| *Mean age at RT (years) [IQR]* | 54.4 [16.4, 45.6–62.0] |
| *Mean intercranial volume (L) [IQR]* | 1.50 [0.166, 1.40–1.56] |
| *Tumor-containing hemisphere (R/L) / lobe* | |
| Right | 5 (41.7) |
| Frontal | 2 |
| Temporal | 1 |
| Frontal/temporal | 1 |
| Parietal/occipital | 1 |
| Left | 7 (58.3) |
| Hippocampal | 1 |
| Occipital | 1 |
| Parietal | 1 |
| Temporal | 1 |
| Frontal/parietal | 1 |
| Frontal/temporal | 1 |
| Parietal/occipital | 1 |
| *Initial surgical procedure* | |
| Resection | 10 (83.3) |
| Biopsy | 2 (16.7) |
| *Tumor grade* | |
| Grade IV | 9 (75.0) |
| Grade III | 2 (16.7) |
| Grade II, GBM markers +* | 1 (8.33) |
| *MGMT promotor status* | |
| Methylated | 9 (75.0) |
| Unmethylated | 2 (16.7) |
| Missing | 1 (8.33) |

Table with demographic and clinical information of study population.

*Based on the diagnostic algorithm for the classification of gliomas in adults by the European council of neuro-oncology, this glioma was classified as a high grade glioma. Abbreviations: GBM = glioblastoma; MGMT, O(6)-methylguanine-DNA methyltransferase; RT = Radiotherapy.

parietal gyrus), temporal (superior temporal) and occipital (calcarine, precuneus, cuneus and superior occipital gyrus) lobe (Fig 2C). Three small clusters for white matter volume loss were found, mostly located in the corpus callosum, putamen, pallidum, insula, cingulate gyri (middle and posterior gyrus) and parietal (supramarginal and postcentral gyrus) and temporal (superior and middle temporal gyrus) lobe when comparing post-RT images with the images after six cycles of TMZ (Fig 2D). No significant clusters were found for volumetric difference of white matter when comparing the images after three cycles and after six cycles of TMZ.

## Gray matter analysis

Voxel-wise comparison of gray matter volume revealed two significant clusters indicating gray matter volume loss when comparing images after three cycles and six cycles of TMZ (Table 3,

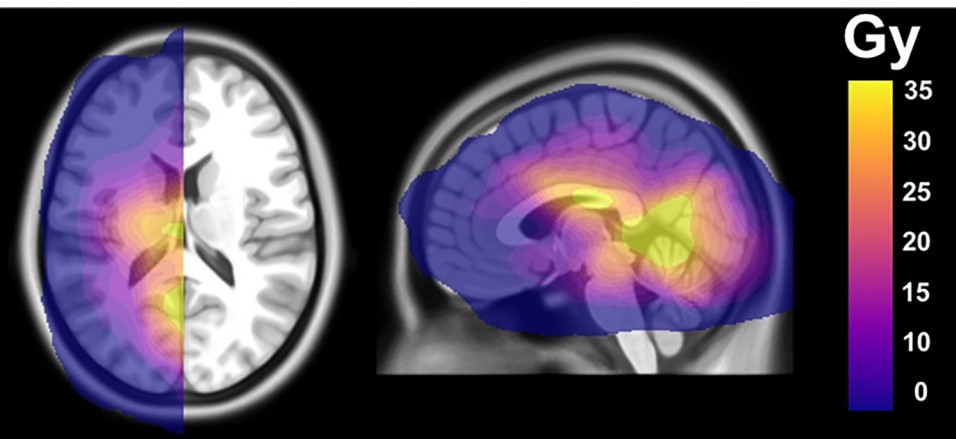

**Fig 1. Mean radiotherapy dose map of the study population.** Overview of the mean radiotherapy dose distribution in the tumor-free hemisphere for the entire patient population, displayed on an T1 weighted MNI brain. Areas that received the highest RT dose (>20 Gy) included the parietal and occipital lobes and thalamus region. The temporal and frontal lobes received a lower dose (10–20 Gy) of RT.

Fig 3). One cluster was located within the putamen and pallidum and the other was located in the cerebellum. No other significant clusters for volumetric difference of gray matter were found for any of the other time-group comparisons (Table 3).

**Table 2. White matter volume differences per timepoint comparison.**

| Brain region | Cluster-level | | Peak-level | | MNI coordinates | | |
|---|---|---|---|---|---|---|---|
| | $p_{FWEc}$ | $K_E$ | T | $Z_E$ | x | y | z |
| *PreRT versus post-RT (4.02* | | | | | | | |
| No significant clusters | - | - | | | - | - | - |
| *PreRT versus after 3 cycles of Temozolomide (4.02)* | | | | | | | |
| Calcarine sulcus | <0.001 | 13062 | 5.84 | 3.86 | 24 | -59 | 11 |
| *PreRT versus after 6 cycles of Temozolomide (4.30)* | | | | | | | |
| Supramarginal gyrus | <0.001 | 27247 | 10.97 | 4.79 | 50 | -39 | 26 |
| Corpus callosum | <0.001 | 2157 | 7.01 | 4.00 | 8 | -27 | 17 |
| *Post-RT versus after 3 cycles of Temozolomide (4.02)* | | | | | | | |
| Midcingulate cortex | <0.001 | 5678 | 11.18 | 5.17 | 8 | -38 | 33 |
| Putamen | <0.001 | 1176 | 6.95 | 4.22 | 32 | -8 | 5 |
| Hippocampus | 0.007 | 653 | 6.63 | 4.13 | 38 | -15 | -11 |
| Supramarginal gyrus | 0.005 | 692 | 6.12 | 3.96 | 59 | -20 | 30 |
| Superior temporal gyrus | 0.004 | 708 | 5.61 | 3.78 | 48 | 17 | 2 |
| *Post-RT versus after 6 cycles of Temozolomide (4.30)* | | | | | | | |
| Postcentral gyrus | <0.001 | 1568 | 7.10 | 4.03 | 27 | -38 | 57 |
| Thalamus | 0.003 | 551 | 7.71 | 4.30 | 12 | -12 | 9 |
| Corpus callosum | 0.014 | 770 | 8.28 | 4.18 | 6 | -26 | 17 |
| *After 3 cycles of Temozolomide versus after 6 cycles of Temozolomide (4.30)* | | | | | | | |
| No significant clusters | - | - | - | - | - | - | - |

Overview of the significant clusters for white matter volume loss of time-group comparisons. Brain regions are reported for the peak levels of each cluster according to the automated anatomical labeling template. Abbreviations: pFWEc = p-value after familywise error correction, KE = cluster size, MNI = Montreal Neurological Institute, RT = Radiotherapy and ZE = Z-value of cluster.

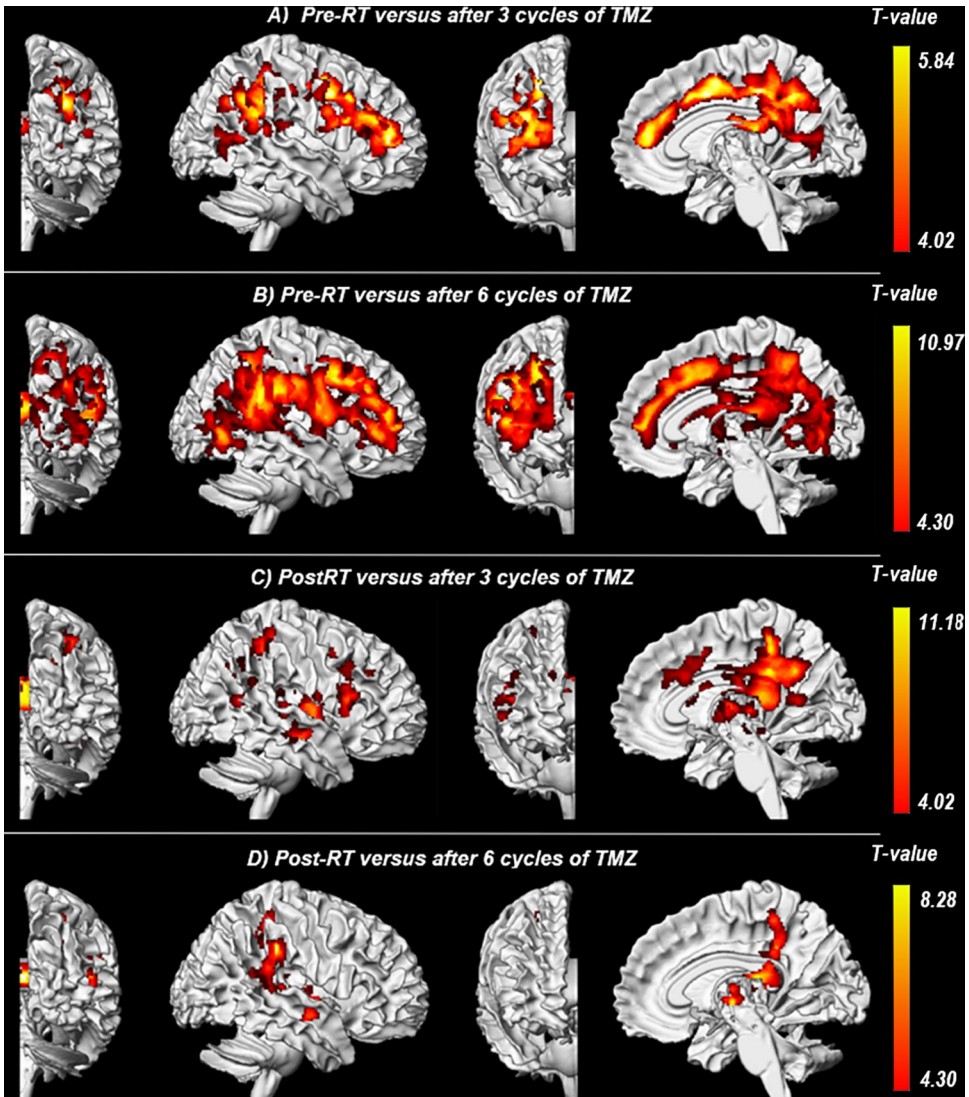

**Fig 2. Significant clusters for white matter volume loss per timepoint comparison.** Overview of the significant clusters for white matter volume loss of time-group comparisons. Different views of the tumor-free hemisphere are displayed. The local cluster color indicates the voxel t-value from the lowest value (red) to the highest (yellow) value. C and D indicate that volume loss is partially persistent over time. Abbreviations: RT = radiotherapy and TMZ = temozolomide.

## Discussion

In this retrospective longitudinal study, VBM analysis demonstrated treatment-induced volumetric changes over time in gray and white matter of the tumor-free hemisphere of 12 HGG patients. We detected diffuse loss of white matter volume, mainly throughout the frontal and parietal lobe of the tumor-free hemisphere. Significant loss of white matter was first noticed after three cycles of TMZ, approximately 16 weeks post-RT, mainly throughout the frontal and parietal lobe. Similar areas of white matter volume loss were observed after the completion of the Stupp protocol. However, no significant loss of white matter volume was observed between pre-RT and the first post-RT follow-up timepoint, indicating a delayed effect.

**Table 3. Gray matter volume differences per timepoint comparison.**

| Brain region | Cluster-level | | Peak-level | | MNI coordinates | | |
|---|---|---|---|---|---|---|---|
| | $p_{FWEc}$ | $K_E$ | T | $Z_E$ | x | y | z |
| *PreRT versus post-RT* | | | | | | | |
| No significant clusters | - | - | - | - | - | - | - |
| *PreRT versus after 3 cycles of Temozolomide* | | | | | | | |
| No significant clusters | - | - | - | - | - | - | - |
| *PreRT versus after 6 cycles of Temozolomide* | | | | | | | |
| No significant clusters | - | - | - | - | - | - | - |
| *Post-RT versus after 3 cycles of Temozolomide* | | | | | | | |
| No significant clusters | - | - | - | - | - | - | - |
| *Post-RT versus after 6 cycles of Temozolomide* | | | | | | | |
| No significant clusters | - | - | - | - | - | - | - |
| *After 3 cycles of Temozolomide versus after 6 cycles of Temozolomide* | | | | | | | |
| Globus pallidus | 0.036 | 459 | 9.74 | 4.59 | 27 | -9 | 2 |
| Cerebellar Crus I | 0.025 | 506 | 8.85 | 4.42 | 23 | -84 | -29 |

Overview of the significant clusters for gray matter volume loss of time-group comparisons. Brain regions are reported for the peak levels of each cluster according to the automated anatomical labeling template. Abbreviations: $p_{FWEc}$ = p-value after familywise error correction, $K_E$ = cluster size, MNI = Montreal Neurological Institute, RT = Radiotherapy and $Z_E$ = Z-value of cluster.

Early-delayed RT damage, occurring weeks to months after RT, leading to white matter volume loss is a multifactorial process and most likely due to transient demyelination, blood-brain barrier disruption and neuroinflammation [22–24]. This RT-induced decrease in white matter volume has also been previously demonstrated [25, 26]. Our study, however, offers a unique group-based analysis with identification of common white matter areas most affected by multi-modality treatment. Previous studies using diffusion tensor imaging (DTI) have found similar results. However, these studies only provide a limited insight in the localization of white matter volume loss within the brain. One study by Conner et al found that normal appearing white matter receiving >30 Gy showed significant diffusion changes four to six months after RT [6]. In line with our results, no significant diffusion changes were observed one month after RT. Another DTI study analyzed multiple distinct regions of interest within the normal appearing white matter of glioblastoma patients receiving 60 Gy fractionated cranial RT [7]. Similar to our findings, it was established that the white matter of the corpus

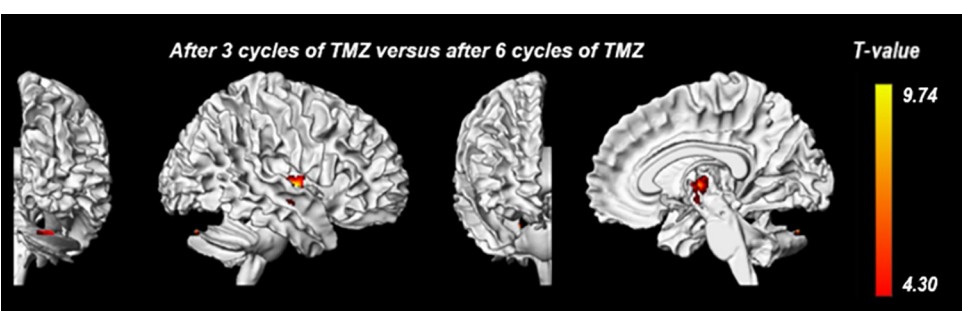

**Fig 3. Significant clusters for gray matter volume loss per timepoint comparison.** Overview of the significant clusters for gray matter volume loss of time-group comparisons. Different views of the tumor-free hemisphere are displayed. The local color of the cluster indicates the voxel t-value from the lowest value (red) to the highest (yellow) value. Abbreviations: TMZ = temozolomide.

collosum, anterior cingulate and fornix were most susceptible to injury as a result of high dose fractionated cranial RT. A study combining VBM analysis and DTI among 14 glioblastoma patients, however, found no significant volume changes of white matter in the tumor-free hemisphere during treatment [27]. However, this study evaluated white matter as a whole, rather than the cluster-based analysis which our study employed. It should also be noted that VBM measured volume changes cannot directly be compared to DTI-based changes, with the latter technique demonstrating different macro/micro changes than VBM analysis.

No significant loss of white matter volume was noticed between the second (after three cycles of TMZ) and third (after completion of the Stupp protocol) post-RT timepoints, suggesting that RT-induced volume loss was not a progressive process in our cohort. Furthermore, these brain areas showing a delayed loss of white matter volume from baseline were comparable for both the pre-RT and the first post-RT baseline timepoints. However, when compared to the pre-RT baseline, the found clusters were more compact in size than compared to the first post-RT timepoint. Although we did not find any significant clusters for white matter volume increase between any timepoint comparison, these findings may indicate that white matter partially recovers. Partial recovery of RT-induced white matter damage has also been described by an earlier DTI study [28]. In this study among eight patients treated with RT, transient white matter changes of the tumor-free hemisphere were demonstrated after three to five months post-RT which recovered at later timepoints of six to nine months and 10–12 months after completion of RT.

Diffuse loss of gray matter and cortical thinning after high-dose RT has been previously described [29–31]. A study among 15 HGG patients found a dose-dependent thinning of the cortical surface one year after RT [32]. Similar findings were described in studies by Seibert et al and Nagtegaal et al [33, 34]. We did not observe any significant gray matter volume difference at any of the post-RT follow-up timepoints compared to the pre-RT baseline. However, two small areas of gray matter volume loss in the globus pallidus and cerebellar crus, were found when comparing images of the third and fourth post-RT follow-up timepoint. Volume decline of deep subcortical gray matter structures, including the globus pallidus, have also been previously demonstrated by another study among 31 glioma patients [31]. It might be possible that gray matter changes due to high-dose RT are a delayed process. However, the end point of this study was after completion of the Stupp protocol (6 months post-RT), which does not allow measurement of delayed effects. Moreover, the aforementioned studies measured cortical thinning across the entire brain whereas we studied the tumor-free hemisphere only, possibly explaining the differences in results.

Comparing our VBM results with our mean RT dose map indicates that regions of white matter volume loss within the thalamus, caudate nucleus, middle and posterior cingulate cortex, corpus callosum and the parietal, temporal and occipital lobe grossly overlap with areas exposed to the highest RT dose (20–30 Gy). These results suggest that, although RT and chemotherapy are both neurotoxic, RT has a strong relationship with the observed white matter volume loss. However, more frontally located areas such as the frontal lobe, anterior cingulate cortex and insula received a lower RT dose but also experienced white matter volume loss. Similar findings of widespread morphological changes in the entire brain were also observed in a VBM study by Nagtegaal et al [35], which utilized pre-RT and post-RT MRI scans of 28 glioma patients. Furthermore, they established that volume changes were dose-dependent. Within our study, despite heterogenicity of the RT field across the study population, group results still yielded common areas affected by RT, indicating a strong relationship between any RT dose and the vulnerability of these areas to RT. This potentially indicates enhanced radiosensitivity of aforementioned more frontally located brain areas.

The largest limitation of this study was the limited sample size, which was a direct consequence of the high requirements of the inclusion and exclusion criteria. To improve the reliability of our results, the analysis was performed on 3D T1-weighted MRI sequences for adequate volume calculations and over multiple timepoints. Additional larger VBM studies are therefore necessary to validate our findings. The small sample size also hindered a voxel threshold with familywise error correction set at <0.05. Future studies should aim to include a larger patient population so that familywise error correction can be utilized at voxel level. Secondly, abnormalities such as tumors within the brain are not reliably segmented into the correct tissue class, making analysis in areas in close proximity to the tumor unreliable. Hence, our analysis was restricted to the tumor-free hemisphere of the brain. Thirdly, the distribution of radiation exposure was not identical across the study population due to difference in tumor location. The unavailability of RT dose maps hindered a direct voxel-wise comparison between RT dose and white or gray matter volume changes on the individual level. Future studies should aim to incorporate RT dose maps into the analysis. However, despite individual differences in RT dose maps, our group-based results still showed common areas affected by standard treatment, probably due to the vulnerability of these areas and their strong relation with any RT dose. Furthermore, the MGMT promotor status was not known for all patients, and its impact of our results was not tested. MGMT methylated tumors are more sensitive to alkalizing chemotherapy and might therefore respond differently than unmethylated tumors in terms of volume change. Finally, unfortunately no neurocognitive evaluation of the HGG patients was available. Future prospective studies should include neurocognitive testing to relate imaging findings to the clinical setting and the impact of patient neurocognitive functioning.

## Conclusion

This study demonstrated diffuse decreases in white matter volume of the tumor-free hemisphere in HGG patients after standard multi-modality treatment according to the Stupp-protocol. These volume changes occurred in the early-delayed phase and were not progressive in nature, suggesting partial recovery. These findings provide an insight into the mechanism of treatment-induced damage to the tumor-free brain tissue in treated HGG patients and suggest that caution should be taken with RT target volume dose planning to minimize white matter injury.

## Author Contributions

**Conceptualization:** Bart R. J. van Dijken, Hiska L. van der Weide, Roelien H. Enting, Anouk van der Hoorn.

**Formal analysis:** Jesse D. de Groot, Bart R. J. van Dijken, Anouk van der Hoorn.

**Investigation:** Bart R. J. van Dijken, Roelien H. Enting, Anouk van der Hoorn.

**Methodology:** Jesse D. de Groot, Bart R. J. van Dijken, Hiska L. van der Weide, Roelien H. Enting, Anouk van der Hoorn.

**Project administration:** Jesse D. de Groot, Bart R. J. van Dijken, Anouk van der Hoorn.

**Software:** Jesse D. de Groot, Hiska L. van der Weide, Anouk van der Hoorn.

**Supervision:** Bart R. J. van Dijken, Hiska L. van der Weide, Roelien H. Enting, Anouk van der Hoorn.

**Visualization:** Jesse D. de Groot, Bart R. J. van Dijken, Hiska L. van der Weide, Anouk van der Hoorn.

**Writing – original draft:** Jesse D. de Groot, Bart R. J. van Dijken, Anouk van der Hoorn.

**Writing – review & editing:** Jesse D. de Groot, Bart R. J. van Dijken, Hiska L. van der Weide, Roelien H. Enting, Anouk van der Hoorn.

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
