## [Decision Letter · Decision Letter 0]

4 Jan 2023

PONE-D-22-25162Voxel based morphometry-detected white matter volume loss after multi-modality treatment in high grade glioma patientsPLOS ONE

Dear Dr. Van Dijken,

Thank you for submitting your manuscript to PLOS ONE. After careful consideration, we feel that it has merit but does not fully meet PLOS ONE’s publication criteria as it currently stands. Therefore, we invite you to submit a revised version of the manuscript that addresses the points raised during the review process.

We look forward to receiving your revised manuscript.

Kind regards,

Kevin Camphausen

Academic Editor

PLOS ONE

and https://journals.plos.org/plosone/s/file?id=ba62/PLOSOne_formatting_sample_title_authors_affiliations.pdf.

Reviewers' comments:

Reviewer's Responses to Questions

**Comments to the Author**

1. Is the manuscript technically sound, and do the data support the conclusions?

Reviewer #1: No

Reviewer #2: Partly

2. Has the statistical analysis been performed appropriately and rigorously? 

Reviewer #1: No

Reviewer #2: No

3. Have the authors made all data underlying the findings in their manuscript fully available?

Reviewer #1: No

Reviewer #2: No

4. Is the manuscript presented in an intelligible fashion and written in standard English?

Reviewer #1: Yes

Reviewer #2: Yes

5. Review Comments to the Author

Reviewer #1: The manuscript aims to tackle an important question, that of white matter volume loss after multi-modality

treatment in high grade glioma patients. They aim to do this by analyzing the impact of radiation therapy dose distribution on areas of the brain that are not believed to be directly involved with tumor.

The authors find diffuse loss of white matter volume, mainly throughout the frontal and parietal lobe of the tumor-free hemisphere and significant loss of white matter after three cycles of TMZ, approximately 16 weeks post-RT, throughout the frontal and parietal lobe but no significant loss of white matter volume between

pre-RT and the first post-RT follow-up timepoint concluding a possible delayed effect. This has been described and the authors note this in the discussion.

These findings are interesting but are likely far more nuanced.

The most significant aspects that if added would enrich the value of the findings, are the addition of information with respect to patient, disease and radiation therapy characteristics. The authors acknowledge that both RT and chemo are neurotoxic. The nuance here is that if patients are MGMT methylated (status not reported) and/or have more significant tumor burden ( GTV, CTV not reported) or had a larger PTV ( not reported) or more significant T2 FLAIR abnormality (was this treated? to 60Gy, to 46 Gy?), were treatments single phase one vol to 60 Gy or two phase with differential margin for T1 gad vs T2 FLAIR ( not reported), all of these aspects would affect dose and volume and dose spillage to the contralat hemisphere. Did the radiation oncologist(s) allow for PTV spillage into the contralateral hemisphere? ( some do and some do not for HGG).

If MGMT methylated, response to chemo in chemo sensitive cells will likely also have a role in white matter changes, particularly if larger tumor or larger margins/PTV especially considering the infiltrative nature of the disease. There is no analysis of the RT dose volume histogram which the authors mention as a limitation but this is a crucial aspect to define the dose to volume relationships to be able to generate conclusions. All the patients were treated VMAT but this does not tell the whole story of the dose distribution and is too broad of a common denominator to assume that given more detailed treatment planning information the conclusion would remain the same.

It is also not clear how CT images were manually divided to acquire RT dose distribution data of the total cerebrum and of each hemisphere. Why not use the RT treatment planning system functionality for this? and report mean and max doses in relationship to the organs at risk, in this case the vulnerable areas of the brain, that the paper wishes to explore dose/white matter change relationships to.

in its current format, the methodology is interesting but the findings are not detailed enough to be more that descriptive although with additional information, there is great potential.

Reviewer #2: The manuscript by Jesse D. de Groot et al, titled as “Voxel based morphometry-detected white matter volume loss after multi-modality treatment in high grade glioma patients” try to evaluated the effect of standard treatment with RT and chemotherapy on gray and white matter volume of the tumor-free hemisphere of patients with unilateral high-grade gliomas using longitudinal voxel based morphometry analysis with SPM.

In the introduction section, I’d suggest to add additional references with longitudinal VBM-based analysis, and not to limit with Parkinson disease. Moreover, it would be more adequate adding references for longitudinal brain changes, including functional, structural and morphometric changes, in patients with brain tumours. There are already several studies reporting longitudinal changes (such as VBM, DTI, perfusion MRI), including pre- and post-treatment in patients with brain tumours (Hye In Lee et al 2022; Hu et al, 2020; Cayuela N et al 2019; Fathallah-Shaykh et al 2019).

From the prospective of similar studies, it should be better introduce what has been done by previous research and the notch that will be added by current study.

In the Material and Method section, RT planning should be moved after MRI acquisition protocol. Missing temporal windows of serial MRI acquisitions.

Authors mentioned in the statistical analysis that paired t tests were applied to compare grey and white volume between two distinct time points. I guess, voxel-wise repeated measure ANOVA was not run. Please, explain the selection of the statistical approach.

Please, explain also why the voxel level PFWE < 0.05 has not been chosen for the level of significance. This fact should be properly addressed and should be also added in the limitation section.

In the Table 2, the voxel-level information should be added with T-statistics

Missing colour-bar for the figure 2 and 3

How the authors address the RT-related changes vs plasticity-related changes in the brain. Does serial neurocognitive assessment available for the patients?

In the discussion section, it should be mentioned that white matter VBM compared to DTI-based analysis are not the same, and shows different macro/micro changes.

It might be interesting to address the interplay between received RT-dose and white and grey matter changes either voxel-wise or by extracting mean volume for each subject from the significant clusters.

Some typos – “gray matter” instead of “grey matter”

6. PLOS authors have the option to publish the peer review history of their article (what does this mean?). If published, this will include your full peer review and any attached files.

Reviewer #1: No

Reviewer #2: No

---

## [Author Response · Author response to Decision Letter 0]

14 Feb 2023

Author response to reviewers' comments 

Reviewer #1

Comment 1: 

The manuscript aims to tackle an important question, that of white matter volume loss after multi-modality treatment in high grade glioma patients. They aim to do this by analyzing the impact of radiation therapy dose distribution on areas of the brain that are not believed to be directly involved with tumor.

The authors find diffuse loss of white matter volume, mainly throughout the frontal and parietal lobe of the tumor-free hemisphere and significant loss of white matter after three cycles of TMZ, approximately 16 weeks post-RT, throughout the frontal and parietal lobe but no significant loss of white matter volume between pre-RT and the first post-RT follow-up timepoint concluding a possible delayed effect. This has been described and the authors note this in the discussion. These findings are interesting but are likely far more nuanced.

Answer:

We would like to thank the reviewer for critically reading our manuscript and providing the suggestions made below. We agree with the reviewer that treatment-induced brain damage in glioma patients is an important topic. In this study, we aimed to provide an insight into one of the possible mechanisms of treatment-induced damage to the tumor-free brain tissue by evaluating white matter volume changes over time in treated high-grade glioma patients.

Comment 2:

The most significant aspects that if added would enrich the value of the findings, are the addition of information with respect to patient, disease and radiation therapy characteristics. The authors acknowledge that both RT and chemo are neurotoxic. The nuance here is that if patients are MGMT methylated (status not reported) and/or have more significant tumor burden ( GTV, CTV not reported) or had a larger PTV ( not reported) or more significant T2 FLAIR abnormality (was this treated? to 60Gy, to 46 Gy?), were treatments single phase one vol to 60 Gy or two phase with differential margin for T1 gad vs T2 FLAIR ( not reported), all of these aspects would affect dose and volume and dose spillage to the contralat hemisphere. Did the radiation oncologist(s) allow for PTV spillage into the contralateral hemisphere? (some do and some do not for HGG).

If MGMT methylated, response to chemo in chemo sensitive cells will likely also have a role in white matter changes, particularly if larger tumor or larger margins/PTV especially considering the infiltrative nature of the disease. There is no analysis of the RT dose volume histogram which the authors mention as a limitation but this is a crucial aspect to define the dose to volume relationships to be able to generate conclusions. All the patients were treated VMAT but this does not tell the whole story of the dose distribution and is too broad of a common denominator to assume that given more detailed treatment planning information the conclusion would remain the same.

Answer: 

The reviewer correctly discussed several significant clinical and radiotherapeutic parameters that were not mentioned in our manuscript. Unfortunately, not all radiotherapy data were available for our analysis, which we indeed mention as a limitation in the discussion section of our manuscript as noted by the reviewer. However, in line with the reviewer’s suggestions, we have added the following additional information to our manuscript: 

All patients received a total radiation dose of 60 Gy, administered as 30 fractions of 2 Gy daily during 6 weeks. The clinical target volume (CTV) consisted of the resection cavity and/or residual tumor plus a 1.5 cm margin without dose spillage to the contralateral hemisphere (page 6). 

As suggested by the reviewer, we have checked the MGMT status of our patients and found the following results: 3 patients with missing MGMT status, 3 patients were MGMT unmethylated and 6 patients were MGMT methylated. We included these numbers to table 1 and added the following sentences to the results section: The O(6)-methylguanine-DNA methyltransferase (MGMT) promotor status was known in 9/12 (75%) patients. MGMT was methylated in 6 patients (50%) and unmethylated in 3 patients (25%) (page 8).

We also mentioned the MGMT status as possible limitation in the discussion section: Furthermore, the MGMT promotor status was not known for all patients, and its impact of our results was not tested. MGMT methylated tumors are more sensitive to alkalizing chemotherapy and might therefore respond differently than unmethylated tumors in terms of volume change (page 16). 

Comment 3:

It is also not clear how CT images were manually divided to acquire RT dose distribution data of the total cerebrum and of each hemisphere. Why not use the RT treatment planning system functionality for this? and report mean and max doses in relationship to the organs at risk, in this case the vulnerable areas of the brain, that the paper wishes to explore dose/white matter change relationships to.

Answer:

The radiotherapy planning software which was utilized was only able to automatically outline of the whole cerebrum and cerebellum. Therefore, the ipsilateral hemisphere of the brain was manually captured within a box and was subtracted from the total brain volume to retrieve information on the contralateral hemisphere.

Comment 4:

In its current format, the methodology is interesting but the findings are not detailed enough to be more that descriptive although with additional information, there is great potential.

Answer: 

Once again, we would like to thank the reviewer for his/her compliments to our work and the suggestions made. We have added more detailed information to our method section in line with the reviewer’s comments and we feel that these suggestions have indeed strengthened our manuscript.

 

Reviewer #2 

Comment 1:

The manuscript by Jesse D. de Groot et al, titled as “Voxel based morphometry-detected white matter volume loss after multi-modality treatment in high grade glioma patients” try to evaluated the effect of standard treatment with RT and chemotherapy on gray and white matter volume of the tumor-free hemisphere of patients with unilateral high-grade gliomas using longitudinal voxel based morphometry analysis with SPM.

Answer:

We thank the reviewer for the constructive feedback presented below, after critically appraising our work. We have addressed the reviewer’s suggestions and added additional information and clarification to our manuscript. 

Comment 2:

In the introduction section, I’d suggest to add additional references with longitudinal VBM-based analysis, and not to limit with Parkinson disease. Moreover, it would be more adequate adding references for longitudinal brain changes, including functional, structural and morphometric changes, in patients with brain tumours. There are already several studies reporting longitudinal changes (such as VBM, DTI, perfusion MRI), including pre- and post-treatment in patients with brain tumours (Hye In Lee et al 2022; Hu et al, 2020; Cayuela N et al 2019; Fathallah-Shaykh et al 2019). From the prospective of similar studies, it should be better introduce what has been done by previous research and the notch that will be added by current study.

Answer: 

We agree with the reviewer that referring to Parkinson’s disease was distracting and not in line with the objective of our study. We thank the reviewer for providing us with additional references to support our introduction, we have added 4 new references (#10, #14, #15, and #16) to our introduction. We have also replaced the Parkinson reference with the following sentences to be more in line with our study objective and highlight what this study adds to the field:

VBM analysis has previously shown to be very useful in detecting longitudinal volume changes in glioma patients. However if, and to what extent treatment affects healthy contralateral brain tissue during treatment, remains largely unknown (page 4). 

Comment 3:

In the Material and Method section, RT planning should be moved after MRI acquisition protocol. 

Answer: 

We have moved the RT planning paragraph below the MRI acquisition protocol as suggested.

Comment 4:

Authors mentioned in the statistical analysis that paired t tests were applied to compare grey and white volume between two distinct time points. I guess, voxel-wise repeated measure ANOVA was not run. Please, explain the selection of the statistical approach.

Answer:

The review correctly concludes that we did not run voxel-wise repeated measure ANOVA, but used pared t tests instead. Repeated measure ANOVA gives no insight in how brain tissue volume changes occur over time. It only indicates if there is a difference between all timepoints. As the goal of our research was to evaluate what the effect of multimodality treatment is over time, repeated measure ANOVA does not provide the requisite information.

Comment 5:

Please, explain also why the voxel level PFWE < 0.05 has not been chosen for the level of significance. This fact should be properly addressed and should be also added in the limitation section.

Answer: 

Due to our low sample size of only 12 patients (which was a result of our strict inclusion and exclusion criteria to provide uniformity), a voxel threshold with familywise error correction set at < 0.05 would be too aggressive. Therefore, this is one of the important limitations of our study. In future studies, similar VBM-studies should aim to include a larger patient population so that familywise error correction can be utilized at voxel level. We have added the following sentences to the limitation section to further emphasize this limitation: The small sample size also hindered a voxel threshold with familywise error correction set at <0.05. Future studies should aim to include a larger patient population so that familywise error correction can be utilized at voxel level (page 16).

Comment 6:

In the Table 2, the voxel-level information should be added with T-statistics.

Answer: 

We have added this to table 2 as suggested. 

Comment 7:

Missing colour-bar for the figure 2 and 3

Answer: 

We thank the reviewer for pointing this out and have indeed included color bars to figures 2 and 3.

Comment 8: 

How the authors address the RT-related changes vs plasticity-related changes in the brain. 

Answer: 

In this study it was not possible to directly differentiate treatment-induced volume changes from potential plasticity changes. However, our analysis was not only performed for white and gray matter volume loss, but also for potential increases in volumes. There were no significant clusters for white or gray matter increase between any timepoints. Furthermore, our results demonstrated early-delayed white matter volume decreases after treatment, which grossly overlapped with areas that received the highest RT dose, further suggesting treatment-induced effects.

Comment 9:

Does serial neurocognitive assessment available for the patients?

Answer: 

Unfortunately no neurocognitive testing was available for our patient cohort. We agree with the reviewer that the correlation between volume changes and neurocognitive decline, which develops in a large percentage of treated glioma patients, would be very interesting to study further. Therefore, we changed the last two sentences of our limitations paragraph, where we also mention this limitation, to the following:

Finally, unfortunately no neurocognitive evaluation of the HGG patients was available. Future prospective studies should include neurocognitive testing to relate imaging findings to the clinical setting and the impact of patient neurocognitive functioning (page 16).

Comment 10:

In the discussion section, it should be mentioned that white matter VBM compared to DTI-based analysis are not the same, and shows different macro/micro changes.

Answer: 

We thank the reviewer for this point as it is true that VBM measured white matter changes cannot directly be translated to DTI findings and vice versa. In line with this comment we have added the following to the discussion: It should also be noted that VBM measured volume changes cannot directly be compared to DTI-based changes, with the latter technique demonstrating different macro/micro changes than VBM analysis (page 14).

Comment 11:

It might be interesting to address the interplay between received RT-dose and white and grey matter changes either voxel-wise or by extracting mean volume for each subject from the significant clusters.

Answer: 

Comparing our VBM results with our mean RT dose map for the entire patient cohort, the regions of white matter volume loss grossly overlap with areas exposed to the highest RT dose as we point out in the results and discussion. These results suggest that RT has a strong relationship with the observed white matter volume loss. We agree with the reviewer that it would be interesting to test this hypothesis on the individual patient level, by investigating a relation between received RT-dose and voxel-wise analysis of volume changes. Unfortunately, this was not possible with our data. We have added the following sentence in line with this suggestion: The unavailability of RT dose maps hindered a direct voxel-wise comparison between RT dose and white or gray matter volume changes on the individual level (page 16).

Comment 12:

Some typos – “gray matter” instead of “grey matter”

Answer: 

Throughout the manuscript we used U.S. English spelling rather than U.K. English, leading to altered spelling of certain frequently mentioned words as “gray” and “tumor”. We have checked the manuscript thoroughly and believe we consequently spelled these words in line with the U.S. English standard.

---

## [Editor Report · Decision Letter 1]

7 Mar 2023

Voxel based morphometry-detected white matter volume loss after multi-modality treatment in high grade glioma patients

PONE-D-22-25162R1

Dear Dr. Van Dijken,

We’re pleased to inform you that your manuscript has been judged scientifically suitable for publication and will be formally accepted for publication once it meets all outstanding technical requirements.

Kind regards,

Kevin Camphausen

Academic Editor

PLOS ONE

---

## [Editor Report · Acceptance letter]

13 Mar 2023

PONE-D-22-25162R1 

Voxel based morphometry-detected white matter volume loss after multi-modality treatment in high grade glioma patients 

Dear Dr. Van Dijken:

I'm pleased to inform you that your manuscript has been deemed suitable for publication in PLOS ONE. Congratulations! Your manuscript is now with our production department. 

Kind regards, 

on behalf of

Dr. Kevin Camphausen 

Academic Editor

PLOS ONE